# The Potential Regulation of A-to-I RNA Editing on Genes in Parkinson’s Disease

**DOI:** 10.3390/genes14040919

**Published:** 2023-04-15

**Authors:** Sijia Wu, Qiuping Xue, Xinyu Qin, Xiaoming Wu, Pora Kim, Jacqueline Chyr, Xiaobo Zhou, Liyu Huang

**Affiliations:** 1School of Life Science and Technology, Xidian University, Xi’an 710071, China; wusijia@xidian.edu.cn (S.W.);; 2School of Life Sciences and Technology, Xi’an Jiaotong University, Xi’an 710049, China; 3Center for Computational Systems Medicine, School of Biomedical Informatics, The University of Texas Health Science Center at Houston, Houston, TX 77030, USA

**Keywords:** Parkinson’s disease, A-to-I RNA editing, miRNA regulation, miRNA competition, gene expression

## Abstract

Parkinson’s disease (PD) is characterized by dopaminergic neurodegeneration and an abnormal accumulation of α-synuclein aggregates. A number of genetic factors have been shown to increase the risk of PD. Exploring the underlying molecular mechanisms that mediate PD’s transcriptomic diversity can help us understand neurodegenerative pathogenesis. In this study, we identified 9897 A-to-I RNA editing events associated with 6286 genes across 372 PD patients. Of them, 72 RNA editing events altered miRNA binding sites and this may directly affect miRNA regulations of their host genes. However, RNA editing effects on the miRNA regulation of genes are more complex. They can (1) abolish existing miRNA binding sites, which allows miRNAs to regulate other genes; (2) create new miRNA binding sites that may sequester miRNAs from regulating other genes; or (3) occur in the miRNA seed regions and change their targets. The first two processes are also referred to as miRNA competitive binding. In our study, we found 8 RNA editing events that may alter the expression of 1146 other genes via miRNA competition. We also found one RNA editing event that modified a miRNA seed region, which was predicted to disturb the regulation of four genes. Considering the PD-related functions of the affected genes, 25 A-to-I RNA editing biomarkers for PD are proposed, including the 3 editing events in the *EIF2AK2*, *APOL6*, and *miR-4477b* seed regions. These biomarkers may alter the miRNA regulation of 133 PD-related genes. All these analyses reveal the potential mechanisms and regulations of RNA editing in PD pathogenesis.

## 1. Introduction

Parkinson’s disease (PD) is the second most common neurodegenerative disorder [1,2]. It is a complex progressive disease characterized by motor difficulties and non-motor issues, such as tremors, slowness, somnipathy, mood changes, and cognitive problems [3,4]. Its pathogenesis is modulated by a number of genetic factors [5,6,7], as well as age and environmental exposures [8,9]. The first identified genetic factor for PD is the α-synuclein gene, which regulates synaptic vesicle trafficking and neurotransmitter release [10]. Mutations in the α-synuclein gene are associated with PD risk [11]. One regulator of α-synuclein, the highly conserved Argonaute 2 (*AGO2*), is abnormally expressed in PD patients [12]. The inflammation-associated serine-threonine kinase (*EIF2AK2*) regulates α-synuclein by directly phosphorylating the Ser129 residue, linking it to neurodegenerative disorders such as PD [13,14].

Aside from α-synuclein pathology, the selective degeneration of dopaminergic neurons also modulates PD. A recent study has reported a decline of *EEF1A* in the PD-affected brain [15]. This gene is involved in the prevention of dopaminergic neuronal cell death. Other genetic factors include *MCL1* [16,17] and *AIMP2* [18], both of which are related to dopaminergic neuron cell death and α-synuclein aggregates.

There is increasing evidence of blood biomarkers in PD, including circulating the serum neurofilament light chain [19], small noncoding RNAs [20], and cytokines [21]. Studies on the identification of blood biomarkers have found altered cytosine-methylated regions and abnormal gene expression in PD patients compared to healthy controls [22,23]. Differentiating exosomal miRNAs, such as *miR-10a-5p*, *miR-153*, and *mR-409-3p*, can be detected in the bloodstream and can serve as diagnostic biomarkers of PD [24]. Alterations in blood-based RNA transcriptomes can provide valuable insights into PD diagnostics, progression, and mechanisms. Thus, in this study, 372 blood samples of PD patients were collected from the Parkinson’s Progression Markers Initiative consortium (PPMI) [25,26] for the discovery of novel PD biomarkers.

A-to-I RNA editing is an important transcriptional modification carried out by adenosine deaminases acting on RNA (*ADAR*) [27]. This form of transcriptional modification can alter the protein-coding capacity, generate diverse protein isoforms, and regulate RNA and protein expression [28]. Previous studies have reported A-to-I editing in the pathogenesis of neurological and neurodegenerative disorders. For example, deficient A-to-I editing of the Q/R site in *GluA2* is correlated with synapse loss, neurodegeneration, and behavior impairments in Alzheimer’s disease [29]. In brain tissue samples of PD patients, there are significant decreases in A-to-I RNA editing levels and the specificity of mRNA editing sites [30]. Given the increasing evidence of A-to-I RNA editing in neurological and neurodegenerative disorders [30,31,32] and its multiple roles in transcript diversity [33,34], it is important to investigate the potential mechanisms of A-to-I RNA editing biomarkers for PD.

Here, we developed a bioinformatics pipeline to identify A-to-I RNA editing biomarkers and uncover their potential mechanisms and transcriptomic impact in PD. The pipeline is outlined in Figure 1. First, A-to-I RNA editing events and gene expressions are identified from RNAseq data. Then, the editing–gene associations are analyzed using two statistical methods: quantitative trait loci (QTL) analysis and Pearson correlation. Next, potential mechanisms of RNA editing effects on genes are explored. Specifically, the impact of RNA editing on miRNA functions and regulations of genes are investigated [33,34]. On one hand, RNA editing can introduce new or abolish existing miRNA binding sites, which can directly affect miRNA–RNA binding and the regulation of the targeted gene. On the other hand, new miRNA binding sites can sequester miRNAs from regulating other genes. Similarly, the loss of miRNA binding sites can release miRNAs and allow for increased binding and regulation of other genes that share the same binding site [35]. Finally, RNA editing in the seed regions of miRNAs may change their binding affinities and affect a number of targeted genes. Our pipeline thoroughly reveals the impact of RNA editing on miRNA regulation and gene expression in PD patients.

## 2. Materials and Methods

### 2.1. Samples Involved in This Study

To study the effects of RNA editing on gene expression, whole-blood RNAseq data of 372 PD patients and 169 healthy controls were downloaded from the PPMI cohort [25,26]. Corresponding demographic and clinical information (Appendix A) were also downloaded from this consortium and used in this study. They include indexes describing the severity of the disease, such as Hoehn and Yahr staging scale (H&Y stages) [36], α-synuclein observations, concentrations of amyloid β-protein [37], tau protein levels [38], and unified Parkinson’s disease rating scale [39].

### 2.2. A-to-I RNA Editing Detection

The sequencing reads from RNAseq data were first aligned to hg38 reference genome (GENCODE v36) by STAR (v2.7.9a) [40]. Then, RNA editing events were identified using REDItools [41] with default settings (i.e., minimal read coverage, 10; minimal quality score, 30; and minimal mapping quality score, 255). To ensure confident identifications, (1) only known editing sites from REDIportal (December 2020) [42] were considered, (2) SNPs from dbSNP151 [43] were removed, and (3) candidates with low reads (*n* < 3) or low editing frequencies (<0.1) were filtered out. On average, there were 15,104 editing events per individual. The number of editing events was not associated with the disease state (*p* > 0.05). Furthermore, informative A-to-I RNA editing events that were found in more than 50 PD patients and had a standard deviation of higher than 0.05 were kept in our analysis. Their distributions across different genes, regions, and repeats were analyzed by ANNOVAR (Version: 2020-06-07) [44].

### 2.3. Gene Expression Quantification

Using the aligned reads as described in 2.2, gene expression was quantified (Transcripts per million, TPM) by the RSEM software (Version: 1.3.1) [45]. Low-abundance genes with mean expressions of less than 1 TPM were excluded, leaving us with 6288 informative genes. To reveal the potential effects of RNA editing events on genes related to PD pathogenesis, 2303 PD-related genes (Appendix A) were curated from DisGeNET [46], MalaCards [47], phenopedia [48], KEGG database [49], GWAS catalog [50], and one previous study [13].

### 2.4. Correlation Analysis between A-to-I RNA Editing Events and Genes

To study the associations between A-to-I RNA editing events and genes, two different analysis methods were used: quantitative traits locus (QTL) with MatrixeQTL [51] (FDR < 0.05) and Pearson correlation (*p* < 0.05). Age and sex were considered as covariates in the QTL analysis, which improves the identification sensitivity and reduces non-genetic phenotypic variance. For these gene-associated RNA editing events, their locations in genomic regions were analyzed. In total, 1203 RNA editing events were located in the 3′-UTRs of pre-mRNAs or lncRNAs and one RNA editing event was in the miRNA seed region. These events may alter miRNA binding and regulation of genes, especially for PD-related genes.

### 2.5. Analysis of RNA Editing Effects on Genes via Potential miRNA Regulation Mechanisms

RNA editing events in miRNA-binding targets, such as 3′-UTRs of pre-mRNAs or lncRNAs, may directly disturb the expression of the edited genes. To study this, miRNA-binding regions in wild-type and RNA-edited sequences were first detected by TargetScan (v7.2) [52] and miRanda (v3.3) [53]. The gain of the miRNA-binding target was determined when the miRNA–RNA interactions occurred in the RNA-edited sequences but not in the wild-type sequences, as detected by both tools, and vice versa for the loss of the miRNA-binding target. RNA editing events may alter original miRNA binding sites and directly affect the expressions of edited genes. To further elucidate the biological impact and functions of the RNA editing events, the interactions between edited genes and RNAs were extracted from starBase [54] and their enriched pathways were determined by Metascape (Version: 3.5) [55], DAVID (Version: 6.9) [56], and Enrichr [57].

As previously mentioned, RNA editing events in miRNA-binding regions may not only affect the edited genes but also indirectly alter other genes due to miRNA competition. To investigate this, TargetScan and miRanda were also used to detect miRNA-binding regions in genes competing with the edited genes. To clarify, a competing gene is a gene that shares miRNA binding sites with the ones altered in the RNA-edited gene. This includes both lost (abolished) and gained (acquired) binding sites from the A-to-I editing. Similarly, the functional roles of the competing gene-associated RNA editing events were annotated using starBase and pathway enrichment analysis.

Last but not least, RNA editing events can also be located in the seed regions of miRNAs (2–8 nt). The edited seed region can bind to and regulate a different set of genes than the intended genes. Similar procedures were applied to study the RNA editing events in miRNA seed regions, including the identification of downstream RNAs regulated by the edited miRNA and the analysis of their enriched pathways.

## 3. Results

### 3.1. A-to-I RNA Editing Events Are Involved in Parkinson’s Disease via Their Effects on Gene Expressions

Our RNA editing detection pipeline identifies 25,799 informative RNA editing events (Appendix A). Most of these editing events occur in protein-coding genes (93.49%), non-exonic/splicing regions (99.77%), and Alu repeats (91.76%), as shown in Figure 2A. Their distributions are consistent with the preferred locations of RNA editing events reported in previous studies [33,58]. Moreover, there is a lower overall editing frequency in PD patients compared to healthy controls (*p* = 5.92 × 10^−3^), which is consistent with the decreased editing levels found in the brains of PD patients [30]. These analyses reveal the reliability of the identified RNA editing events.

To associate the frequencies of RNA editing events to gene expressions, quantitative traits loci analysis (QTL, FDR < 0.05) and Pearson correlation (*p* < 0.05) were performed. In total, there are 9897 RNA editing events associated with 6286 genes (Appendix A), as shown in the Manhattan plots in Figure 2B. To understand the functions and impact, we conducted pathway enrichment analysis on these editing-associated genes and found that they are enriched in neurodegeneration-related functions (Figure 2C, Appendix A). Thus, RNA editing events may affect PD pathogenesis via their effects on these genes.

Next, we analyzed the distributions of the editing–gene associations. The densities of editing-associated genes (Figure 2D) and editing events (Appendix A) are different. There are more affected genes in the gene-rich regions of chr19 (Chr19:10M−20M) and chr11 (Chr11:60M-70M) (Figure 2D and Appendix A). In addition, the majority of these associations belong to the trans- and cross-chromosomal RNA editing regulations of genes (Figure 2E). The preference of RNA editing events for regulating distal genes urges us to explore the potential mechanism underlying these associations. The whole analysis process can be divided into three parts. First, RNA editing events may abolish or create miRNA-binding sites within the genes and directly alter their expressions (Section 3.2). Second, RNA editing events may release miRNAs to or sequester miRNAs from other genes, indirectly affecting the expressions of these miRNA-competing genes (Section 3.3). Third, RNA editing events may affect miRNA seed regions and then alter their downstream targets (Section 3.4). All these mechanisms are covered in the next sections.

### 3.2. A-to-I RNA Editing Events May Affect miRNA Regulations of Their Host Genes

A-to-I RNA editing can modulate gene expression by altering the miRNA binding sites of the host gene. In total, there are 581 RNA editing events located in their associated genes (Figure 3A). Of them, 72 RNA editing events were predicted to create 11,370 new miRNA-binding sites and eliminate 815 original ones. The QTL and Pearson correlation results for these editing events and gene expressions are presented in Appendix A.

Interestingly, there is a noteworthy RNA editing event in the 3′-UTR of *EIF2AK2* (Chr2:37104057), as shown in Figure 3B. This RNA editing event creates new binding sites for *miR-3622a-3p* and *miR-3622b-3p*, which may increase the degradation of this gene and result in the observed decrease in gene expression (Figure 3C). In fact, the frequency of this RNA editing event is negatively correlated to the expression of *EIF2AK2* (QTL: FDR = 1.39 × 10^−4^, β = −12.60; Pearson correlation: *p* = 4.94 × 10^−4^, R = −0.42). The associated *EIF2AK2* gene is known for its function in promoting PD progression via the phosphorylation of α-synuclein protein [13]. It is also significantly highly expressed in PD (*p* = 0.01) and in more severe PD samples (*p* = 0.03). The Chr2:37104057 RNA editing event is significantly less frequent in PD samples compared to healthy controls (*p* = 0.02) and is negatively associated with tau protein levels (*p* = 0.06, R = −0.24). Thus, this RNA editing event may be the reason for the observed downregulation of *EIF2AK2* and may be considered a potential biomarker for PD.

To further explore the biological impact and downstream effects of this RNA editing event, we examined the role of the edited gene. *EIF2AK2* encodes for a kinase that binds to dsRNA (RNA binding protein) and plays a major role in the regulation of gene transcription [59]. According to *EIF2AK2*–RNA interactions from StarBase [54], RNAInter [60], and RNAct [61], there are 2158 potential *EIF2AK2* targets and 4129 potential non-*EIF2AK2* targets. All these genes are sufficiently expressed in blood with mean expressions of more than one TPM. Of 2158 potential *EIF2AK2* targets, 31.37% have associations with both this editing event and *EIF2AK2* (Appendix A). Of 4129 potential non-*EIF2AK2* targets, 26.45% have associations with both this editing event and *EIF2AK2* (Appendix A). There is an enrichment of *EIF2AK2*-interacted genes associated with the editing event (Fisher test: *p* = 4.54 × 10^−5^). This suggests that *EIF2AK2* editing may regulate these genes via its RNA binding functions. Taking a closer look at the experimentally validated *EIF2AK2*–RNA interactions from starBase, there are 16 protein-coding RNAs significantly correlated to the editing event and *EIF2AK2* (Figure 3D). The 16 editing-associated genes are involved in the neurodegenerative NIK/NF-KB signaling and immune processes (Figure 3E, Appendix A). For example, the NF-KB gene plays a prominent role in dopaminergic neurodegeneration [62]. The upregulation of inflammatory cytokines can initiate a cascade of pro-inflammatory signaling that ultimately results in neurotoxicity related to PD [6].

Among the 16 affected genes, at least 4 genes (*MCL1* [16], *P4HB* [63], *DNAJB6* [64], and *APP* [65]) are known to be implicated in PD. Specifically, *MCL1* has reported its associations with the loss of dopamine neurons, which is one of the reasons for the motor symptoms in PD [16]. In our study, *MCL1* is overexpressed in PD (*p* = 1.87 × 10^−4^) and in more severe PD patients (*p* = 0.02) (Figure 3F). Its coexpressions with the edited gene (*p* = 3.21 × 10^−36^, R = 0.59) and its negative correlations with the editing event (*p* = 1.73 × 10^−3^, R = −0.38) further support their relationships. Therefore, the decrease of the RNA editing event (Chr2:37104057) may upregulate *EIF2AK2* [13], which then affects the expression of at least four PD-related genes (Figure 3G). Based on our findings, RNA editing events in miRNA targets may perturb miRNA degradation functions, leading to abnormal gene expression and the subsequent altered biological pathways.

### 3.3. A-to-I RNA Editing Events May Alter miRNA Competitions between Their Host Genes and Other Genes

RNA editing events in one gene may indirectly affect other genes. Editing events that create new miRNA binding sites may sequester miRNAs from regulating other RNAs. Similarly, editing events that abolish miRNA binding sites may release the miRNAs to regulate other RNAs. The process is also known as editing-mediated miRNA competition. In our study, eight RNA editing events that can alter miRNA competitions between their host genes and 1146 other genes were identified (Figure 4A, Appendix A). These affected genes are enriched in the biological process related to expression regulation and PD, as shown in Figure 4B and Appendix A. Among them, 127 have reported their roles in PD previously (Appendix A). These PD-related genes may be affected by miRNA dysregulations from four RNA editing events.

The editing event of most interest is the one in the Chr22:35661178 position of *APOL6*. It is located in the 3′-UTR and results in the loss of *miR-19* and *miR-4659* binding sites, as shown in Figure 4C. The lost miRNA regulations may directly increase the expression of *APOL6* (QTL: FDR = 1.84 × 10^−4^, β = 8.55; Pearson: *p* = 7.13 × 10^−8^, R = 0.29). This releases the two miRNAs from *APOL6*, allowing them to regulate other genes and potentially decrease the transcription of 47 genes, including *EEF1A1* (QTL: FDR = 2.29 × 10^−3^, β = −609.93; Pearson: *p* = 6.37 × 10^−7^, R = −0.27; Figure 4D) and 8 ribosomal proteins (Figure 4E, Appendix A). The edited gene, *APOL6*, induces a neurodegeneration-related apoptosis process [66,67,68] and is highly expressed in more severe PD samples (*p* = 0.02). One PD-related gene competing with the edited gene is *EEF1A1*. It is known for its pro-survival function in the protection of dopaminergic neurons [15]. The eight ribosomal proteins and the *EEF1A1* translation factor are responsible for the synthesis of proteins, especially PD-related peptides. The dysregulation of these PD-related genes suggests that this editing event at Chr22:35661178 is a potential pathogenic biomarker for PD (Figure 4F). It is also supported by the highly edited frequencies in more severe PD samples characterized by H&Y stages (*p* = 0.04) and tau protein levels (*p* = 0.09, R = 0.09). The analyses in this section provide one explanation for the trans-associations of A-to-I RNA editing events and genes.

### 3.4. A-to-I RNA Editing Events May Modify miRNA Seed Regions to Disturb Their Regulations

Given that miRNA seed regions are highly conserved, a single nucleotide change (e.g., A-to-I RNA editing) can dramatically alter the targeted genes of the miRNAs. In total, we discovered one RNA editing event in the seed region of *miR-4477b* (Chr9:63819627). This change may dysregulate four genes (Appendix A), including one PD-related gene, *AGO2* (Figure 5A).

This RNA editing event alters the nucleotide from A to I in the fourth position of the miRNA seed region and results in the loss of binding on three sites in *AGO2* (Figure 5B). The loss of *miR-4477b*-*AGO2* is associated with an increased expression of *AGO2* (QTL: FDR = 0.02, β = 7.21; Pearson: *p* = 1.63 × 10^−4^, R = 0.26; Figure 5C). The associated gene, *AGO2*, is involved in the production of α-synuclein [12]. The RNA editing event shows a higher frequency in PD samples compared to the healthy controls (*p* = 0.06). In addition, RNA editing in *miR-4477b* is positively correlated with the levels of amyloid β-protein (*p* = 0.05, R = 0.14) and α-synuclein (*p* = 0.09, R = 0.12). This suggests that the RNA editing event at Chr9:63819627 leads to the loss of miRNA regulation on *AGO2* related to PD.

Furthermore, as an RNA binding protein, *AGO2* interacts with 168 protein-coding RNAs, according to starBase. Of them, 46 RNAs are significantly associated with the editing frequency of *miR-4477b* and the expression of *AGO2* (Figure 5D). These associated genes include the highly conserved Argonaut family of proteins, which includes *AGO1* and *AGO2* and plays a role in short-interfering-RNA-mediated gene silencing (Figure 5E). They also include four other PD-related genes: *AIMP2* (Figure 5F), *PSMC3IP*, *RMDN3*, and *NPC1*. These four genes have been reported to correlate with neurodegenerative diseases [69,70], have activation roles in age-dependent dopaminergic neuronal loss [18], and contain regulatory functions in PD-related autophagy [71]. All the above analyses support that RNA editing events in miRNA seed regions affect their regulatory functions on multiple PD-related genes (Figure 5G).

### 3.5. A-to-I RNA Editing Effects on Genes via Disturbing miRNA Regulations in Other Datasets

In our analysis, we discovered a list of RNA binding events that may alter the miRNA regulations of genes (Appendix A). To further support our findings, we expand our analyses to other cohorts. Three datasets were analyzed: (1) a small PD blood dataset (GSE165082), (2) a healthy blood dataset (PPMI), and (3) a large Alzheimer’s disease (AD) dataset. The small PD blood dataset did not contain the three RNA editing events introduced in detail in this study; however, it provides the validation of one RNA editing event (Chr1:160998027; Figure 6A,B and Appendix A). In the healthy blood dataset and AD dataset, there are 1135 significant associations between RNA editing events and genes (Appendix A), which are consistent with the 1300 associations found in our study (Appendix A). The 1135 significant associations include the three RNA editing events in the *EIF2AK2*, *APOL6*, and *miR-4477b* seed regions described in detail (Figure 6C–H). All these analyses support the mechanism that A-to-I RNA editing events alter miRNA binding and dysregulate genes related to the pathogenesis of PD and AD [33]. In conclusion, this study expands the knowledge of RNA editing effects on neurodegeneration.

## 4. Discussion

In recent years, A-to-I RNA editing in PD has attracted the interest of researchers from all over the world [30,72]. Previous studies focused on the differential RNA editing events in coding regions and miRNAs. However, their mechanisms and the potential functions of the editing events in lncRNAs or 3′-UTRs of mRNAs remain largely unclear. In this study, we analyzed 372 PD patients and identified the associations between 9897 A-to-I RNA editing events and 6286 genes. Of the gene-associated RNA editing events, more than 26% were significantly correlated to PD or PD severity (*p* < 0.05; Appendix A). A total of 1224 PD-specific RNA editing events were also significantly associated with 612 PD genes (Appendix A). These RNA editing events may affect the expression of proximal genes (dist ≤ 1 × 10^6^), also known as cis-regulatory effects. They can also regulate the expression of distant genes (dist > 1 × 10^6^), which are considered trans-regulatory effects. Most of the associations were found to be trans-regulatory ones, in which the RNA editing event was significantly associated with a distal gene (Figure 2C).

To further explore the underlying mechanisms of these associations, we analyzed the perturbed miRNA regulations of genes caused by A-to-I RNA editing events. Of all the 624,325 editing–gene pairs, 77.81% are negatively associated. One reason for the reduced expressions is that RNA editing events caused more gained miRNA binding sites than lost sites (Second 3.2) [33,73]. In total, there are 72 RNA editing events that created or eliminated miRNA binding sites and directly affected miRNA regulations of their host genes (Appendix A). Additionally, there are 8 RNA editing events that released or sequestered miRNAs and indirectly resulted in the dysregulation of 1146 other genes (Appendix A). Finally, there is one RNA editing event that modified a miRNA seed region, which was predicted to potentially disturb the regulations of four genes (Appendix A). Of the 73 RNA editing events that altered miRNA binding (Appendix A), 37 events had differential frequencies in PD and in more severe PD samples (*p* < 0.1, Appendix A). Of the 1179 genes affected by editing-mediated miRNA regulatory alterations (Appendix A), 598 genes displayed abnormal expressions in PD (*p* < 0.05; Appendix A) based on PPMI and four GEO datasets (GSE165082, GSE99039, GSE72267, and GSE18838). According to these analyses and the PD-related functions of the affected genes, 25 A-to-I RNA editing biomarkers for PD are proposed, including the 3 editing events in the *EIF2AK2*, *APOL6*, and *miR-4477b* seed regions. These biomarkers may alter the miRNA regulation of 133 PD-related genes.

To increase the reliability of our study, both QTL and Pearson correlation tests were used to identify the associations between A-to-I RNA editing events and genes. The associations were only determined when they all passed the statistical significance thresholds from the two tests. Among the 627,064 significant pairs identified by QTL analysis, 620,840 also passed Pearson correlation tests (*p* < 0.05). Of them, 620,616 editing–gene associations were still significantly correlated after Benjamini–Hochberg corrections (*p*.adjust < 0.05). To understand the mechanisms underlying these associations, two widely used miRNA-binding prediction tools, TargetScan and miRanda, were applied to identify the effects of RNA editing on the alterations of miRNA binding. The altered miRNA–RNA interactions were only kept when they were identified by both tools. All these criteria ensured the accuracy of the editing–gene associations and the miRNA–RNA interactions.

Our study identified potential RNA editing biomarkers for PD and elucidated their effects on downstream genes and pathways. We presented three editing events associated with dysregulated PD-related genes. These A-to-I RNA editing events may also be affected by other factors, such as the main editing enzyme (*ADAR*) and genetic variants [74] (Appendix A). Collectively, DNA mutations, *ADAR* activities, A-to-I RNA editing events, miRNA regulations, and gene expressions can explain the regulatory processes in neurodegenerative pathogenesis.

Additional research can be conducted to further understand RNA editing effects on PD pathogenesis. For example, multiple RNA editing events can affect multiple miRNA–RNA regulations and produce complex outcomes on PD-related genes (Appendix A). A network to analyze the effects of editing modules on PD hub genes is needed. Furthermore, studies are also needed to explain the many editing–gene pairs identified in our study. Of the 624,325 editing–gene pairs, only 1838 associations can be explained by the miRNA–RNA mechanisms presented in this study. A number of gene-associated RNA editing events are located in RNA-binding proteins or their targets (Appendix A). These can affect the regulations of the expression of multiple genes. Future studies can be conducted to fully capture the regulatory effects of the RNA-binding proteins that are affected by RNA editing. These two studies will help further understand RNA editing functions in PD pathogenesis.

## 5. Conclusions

In sum, we systematically analyzed the effect of A-to-I RNA editing events on the miRNA regulations of PD-related genes. In total, we discovered 25 A-to-I RNA editing biomarkers for PD. These biomarkers may alter the miRNA regulation of 133 PD-related genes. Three RNA editing events in the *EIF2AK2*, *APOL6*, and *miR-4477b* seed regions were discussed in detail. Their effects on the downstream genes and pathways related to PD were revealed. Our work helps researchers understand the mechanisms of the abnormal expressions of PD-related genes and the impact of A-to-I editing in neurodegeneration.

## Figures and Tables

**Figure 1 genes-14-00919-f001:**
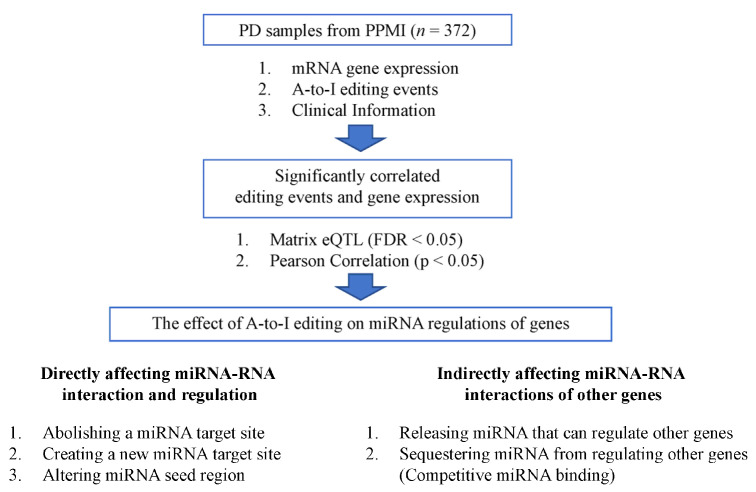
The flowchart of this study.

**Figure 2 genes-14-00919-f002:**
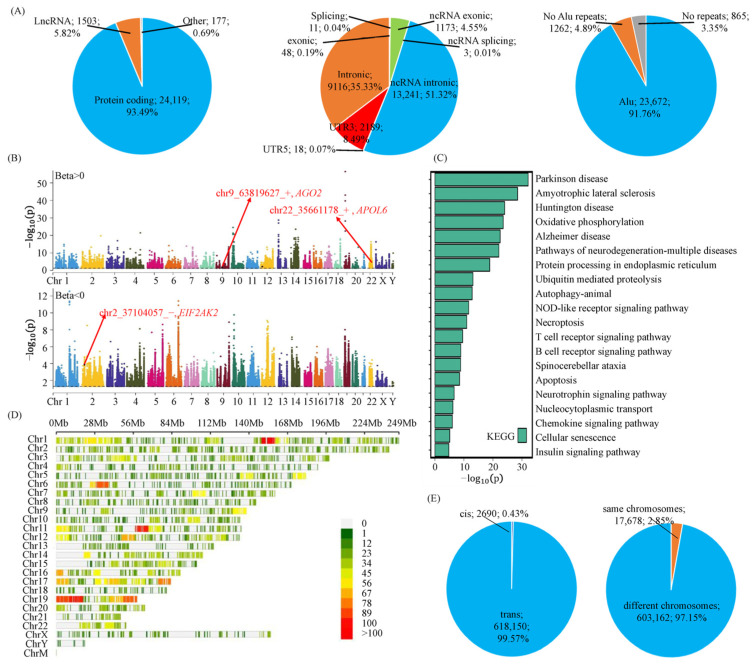
The overview of RNA editing effects on gene expressions. (**A**) The distributions of informative RNA editing events in genes, regions, and repeats. (**B**) Manhattan plots show the effects of 9897 RNA editing events on 6286 genes by QTL (FDR < 0.05) and Pearson correlation analyses (*p* < 0.05). The top and bottom panels present 137,563 positive and 483,277 negative associations, respectively. (**C**) The PD-related enrichment pathways of the editing-associated genes by DAVID. (**D**) The density distributions of editing-associated genes on chromosomes. The color bar shows the number of editing-associated genes in a 10 M bps (base pairs) region. (**E**) The types of editing–gene associations. Most RNA editing events are associated with distal genes (dist > 1 × 10^6^, trans-associations) and genes in different chromosomes.

**Figure 3 genes-14-00919-f003:**
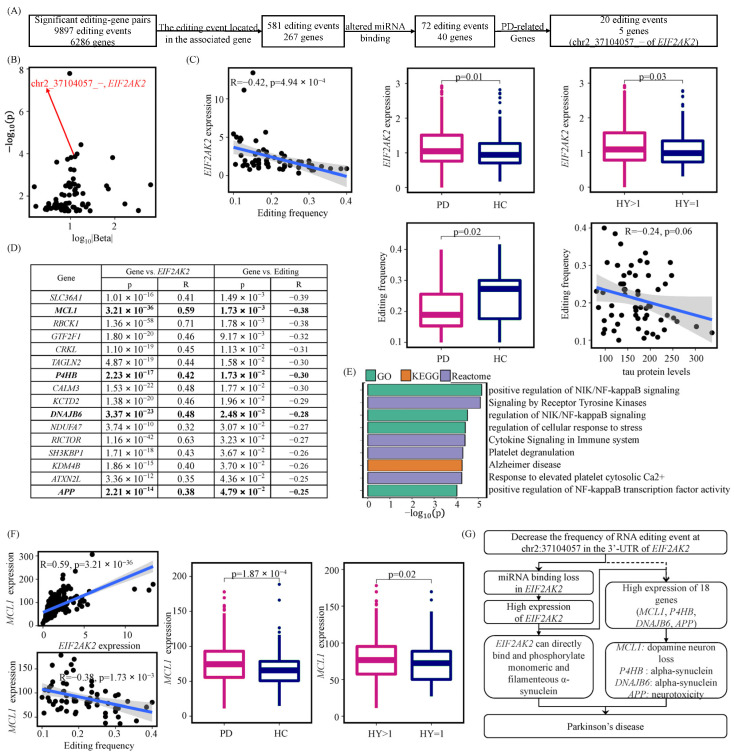
A-to-I RNA editing events may alter miRNA regulations of their host genes. (**A**) Analysis procedures to uncover one mechanism of the associations between RNA editing events and their host genes. (**B**) The scatter plot shows the RNA editing events, which may alter miRNA regulations of their host genes. The editing example in *EIF2AK2* is highlighted. (**C**) This RNA editing event is negatively associated with *EIF2AK2* expression. *EIF2AK2* is overexpressed in PD and in more severe PD samples. The RNA editing event shows lower editing frequencies in PD samples and negative associations with tau protein levels. (**D**) The 16 *EIF2AK2*-interacted genes are significantly associated with *EIF2AK2* and the RNA editing event. Genes known to be implicated in PD are in bold. (**E**) The 16 genes are enriched in neurodegeneration-related processes by Metascape. (**F**) Of them, *MCL1* is overexpressed in PD and in more severe PD samples. (**G**) The analyses above may uncover the potential effects of this RNA editing event on five PD-related genes.

**Figure 4 genes-14-00919-f004:**
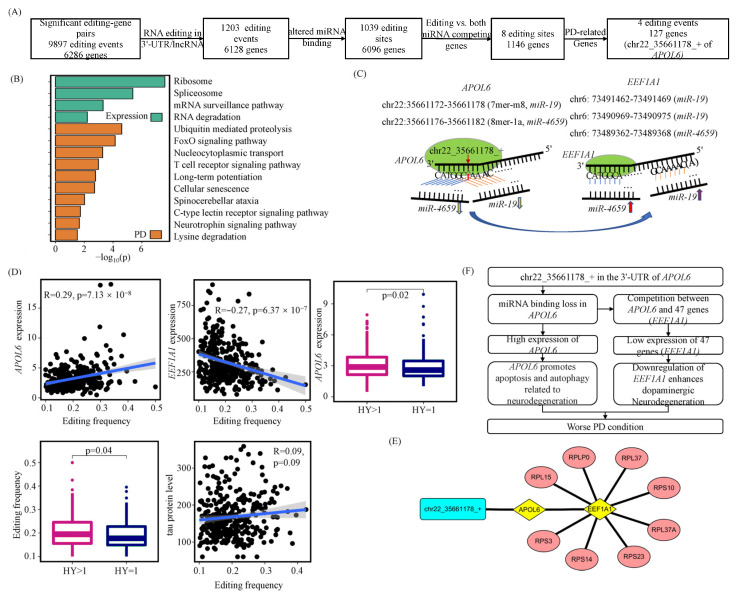
A-to-I RNA editing events may alter miRNA competitions between their host genes and other genes. (**A**) Analysis procedures to uncover one possible mechanism of the trans-associations between RNA editing events and genes. (**B**) The enrichment results of the genes affected by editing-mediated miRNA competitions. (**C**) One RNA editing event in the 3′-UTR of *APOL6* causes the binding loss of miRNAs and releases the miRNAs to regulate *EEF1A1*. (**D**) The frequency of this editing event is positively associated with the expression of *APOL6* and negatively correlated to *EEF1A1*. *APOL6* is highly expressed in severe PD samples. The editing event has higher frequencies in the samples with increased H&Y stages or tau protein levels. (**E**) The editing event may directly result in the overexpressions of *APOL6* and indirectly cause the downregulations of 47 genes including *EEF1A1* and 8 ribosome proteins. (**F**) The potential roles of this RNA editing event related to PD pathogenesis.

**Figure 5 genes-14-00919-f005:**
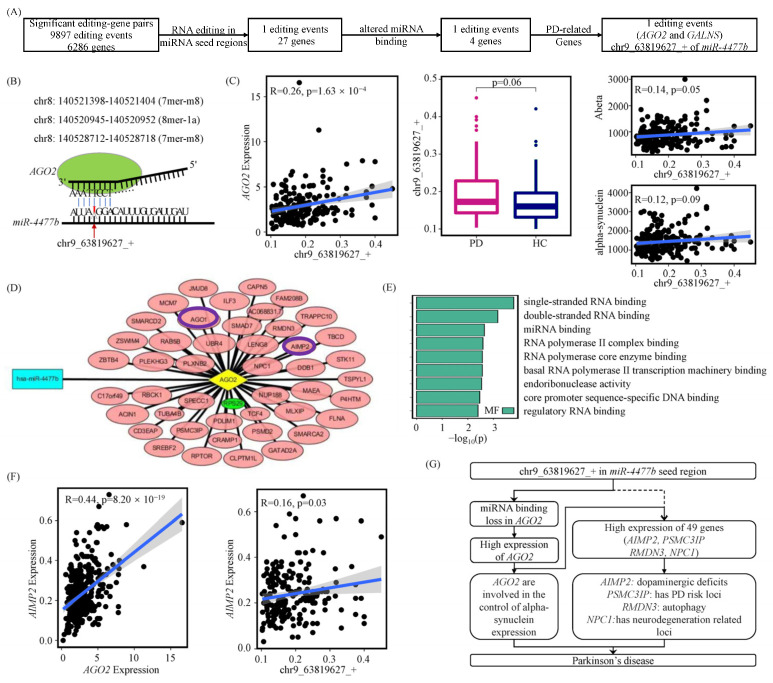
A-to-I RNA editing events may modify miRNA seed regions to disturb their regulations. (**A**) Analysis procedures to uncover the possible mechanisms for the associations between RNA editing events in miRNA seed regions and their targets. The editing example in *miR-4477b* is highlighted. (**B**) This RNA editing event alters the miRNA seed region, thus leading to the loss of miRNA binding on *AGO2*. (**C**) The frequencies of this RNA editing event are positively associated with the expressions of *AGO2*. The editing event shows higher frequencies in PD samples and positive associations with the levels of amyloid β-protein and α-synuclein. (**D**) The 46 *AGO2*-interacted RNAs are significantly associated with *AGO2* and the editing event. The red and green circles describe the positive and negative associations, respectively. (**E**) The associated genes are enriched in the regulations of miRNAs and RNA binding proteins by Enrichr. (**F**) One PD-related gene, *AIMP2*, is positively associated with *AGO2* and the editing event. (**G**) The analyses above may uncover the potential effects of this RNA editing event in miRNA seed region on five PD-related genes.

**Figure 6 genes-14-00919-f006:**
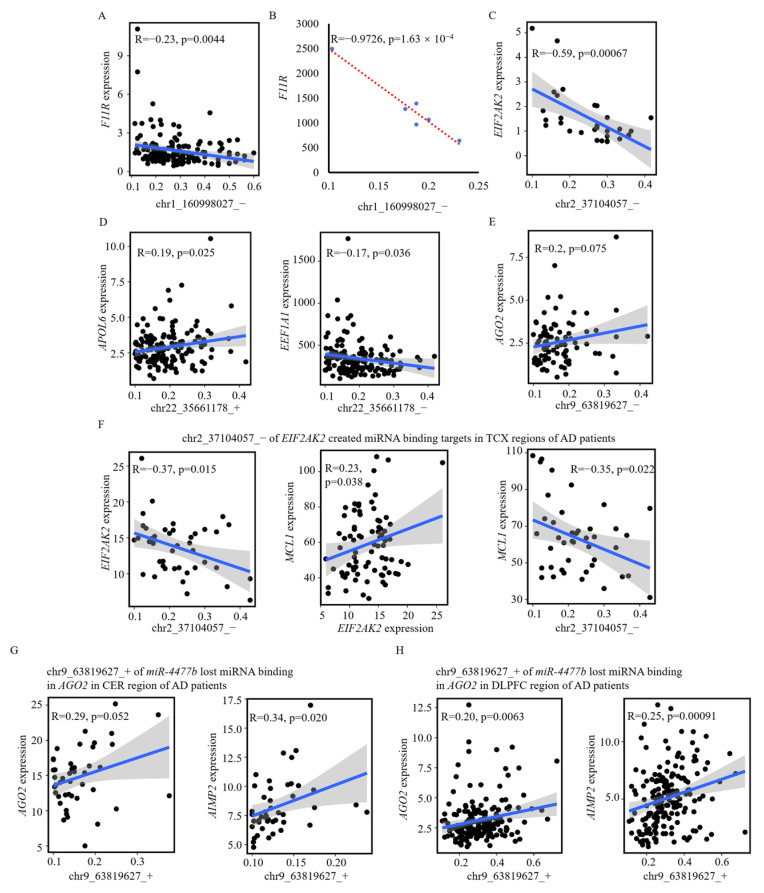
The effects of RNA editing events on genes in other datasets. (**A**,**B**) In blood samples of PD patients from PPMI and GEO (GSE165082), the RNA editing event at Chr1:160998027 shows significant associations with *F11R*. (**C**–**E**) In healthy blood samples, the three RNA editing events discussed in detail also show significant associations with corresponding genes. (**F**) In temporal cortex (TCX) regions of AD patients, the RNA editing event at Chr2:37104057 is significantly associated with *EIF2AK2* and *MCL1*. (**G**,**H**) In cerebellum (CER) and dorsolateral prefrontal cortex (DLPFC) regions of AD patients, the RNA editing event (Chr9:63819627) in the seed region of *miR-4477b* may dysregulate *AGO2* and *AIMP2*.

## Data Availability

The data presented in this study are available in Appendix A here.

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
