# Peer review of "The Potential Regulation of A-to-I RNA Editing on Genes in Parkinson’s Disease"

_genes, 2023, doi:10.3390/genes14040919_

Round 1
Reviewer 1 Report
I would like to star by congratulating the authors on their project. We know that PD progression and severity varies from patient to patient, being important to understand what may underly these clinical differences.
A-to-I RNA editing is a topic of much interest, and the work performed by the authors is noteworthy. It would be important to test it in different cohorts, but possibly for a future study.
I would just recommend an overall English editing, some parts are more difficult to read, or using expression that are not common in medical literature (I completely understand it is not the authors maternal language)
Author Response
I would like to star by congratulating the authors on their project. We know that PD progression and severity varies from patient to patient, being important to understand what may underly these clinical differences. A-to-I RNA editing is a topic of much interest, and the work performed by the authors is noteworthy. It would be important to test it in different cohorts, but possibly for a future study. I would just recommend an overall English editing, some parts are more difficult to read, or using expression that are not common in medical literature (I completely understand it is not the authors maternal language)
>> Thank you for your time and comments. We apologize for the lack of English clarity and the difficulty you experienced when reviewing our manuscript. We have invited Jacqueline Chyr, who is a native English speaker, to fully revise and edit our manuscript. We hope our manuscript is clearer and more coherent.
Reviewer 2 Report
The authors analyze RNA-seq samples for a cohort of Parkinson's disease (PD) patients together with a set of healthy controls to generate a resource of A>I RNA-editing in PD. They use QTLs and correlations to associate editing patterns with the expression of host and putatively impacted target genes. They also look for associations of RNA-editing with the loss/gain of miRNA binding and the downstream impact of this on target genes.
Overall, it is interesting and important to interrogate these kinds of cohort datasets for RNA editing patterns, in order to try to understand the underlying regulatory mechanisms of disease. The current study has merits and will be of interest to readers of MDPI Genes. However, there are some concerns about the rigour of the computational approach (see below), which if addressed could help strengthen the conclusions of the study.
- Major points
- The use of the English language should be improved in this manuscript - it would benefit from a native speaker reading through. Also, since many of the results are of the speculative nature, the word "probably" in quite a few places ought to be softened to "may"/"can" (or perhaps "likely" e.g. line 178).
- Chromosomes 11 and 19 are anyway highly gene-dense chromosomes (chr19 being the most gene-dense in the human genome). Are the editing-associated genes more enriched in these chromosomes than the background density?
- "1,146 miRNA-competing genes (Figure 4A, Table S7). Among these potentially affected genes, 127 are associated with Parkinson's disease"
- Here, the GO term from 4B is only based on those PD genes, so it shows that PD genes have PD-associated GO terms, which, if this is indeed a correct interpretation, isn't surprising. It does not show that PD genes are enriched in this 1,146 miRNA competing gene set. Therefore: Is there a significant overlap between the 127 PD genes and the miRNA competing genes?
- Please expand on this "Of them, 18 RNAs have the relationships with both this editing event and its host gene (Figure 3D)" In 3D what are the bold genes? Furthermore, if you took random sets of 91 (non-EIF2AK2-target) genes, how many would also have this relationship? If it is fewer, then you are supporting the role of EIF2AK2-editing on its RBP function.
- From the resource perspective it would be useful to know how shared are the editing sites across the different people in general? How much editing is present in each individual overall and does this correlate with disease state?
- For the QTL analysis FDR is used - what multiple testing corrections were used for the Pearson's correlation analysis? Furthermore, the strategy for the Pearson analysis is not well explained.
- Cross-checking with the other cohorts is interesting and looks promising - however this part is first presented in the Discussion, perhaps this would be better included as part of the results themselves.
- Minor points
- Since this is blood samples it might be a good idea to frame in the introduction what the connection is between the blood and Parkinson's disease.
- "To investigate this, we also used TargetScan and miRanda to detect miRNA-binding regions in genes competing with the edited genes." - how is a 'competing' gene defined in this study?
- 2D what is the scale of this figure? Please write it in the legend
- "we recognized 72 RNA editing events which may create 11,370 new miRNA-binding targets and eliminate 815" - can one speculate how the presence of the editing events would lead to the creation of so many miRNA binding targets, and also such a high ratio of created vs eliminated?
- 204 - please do not use the word validate, since it is just a correlation. Perhaps use "supported" instead.
- 207 - "we could annotate one potential mechanism" >> "we could speculate one potential mechanism"
- What is "Informative Value" in the context of calling RNA-editing sites?
- The authors mention H&Y stages a few times but don't mention what it is. Please say somewhere so that readers avoid having to search themselves
- Figure 3 - the bright green and red is very distracting (and probably not great for color blind readers).
- What is the measurement/scale in Figure S3?
- Since the focus was on the sites already present in the REDIportal, can the authors speculate on whether more PD-specific sites could be found from the data, which are not in the the portal, and the potential impact of this?
Author Response
The authors analyze RNA-seq samples for a cohort of Parkinson's disease (PD) patients together with a set of healthy controls to generate a resource of A>I RNA-editing in PD. They use QTLs and correlations to associate editing patterns with the expression of host and putatively impacted target genes. They also look for associations of RNA-editing with the loss/gain of miRNA binding and the downstream impact of this on target genes. Overall, it is interesting and important to interrogate these kinds of cohort datasets for RNA editing patterns, in order to try to understand the underlying regulatory mechanisms of disease. The current study has merits and will be of interest to readers of MDPI Genes. However, there are some concerns about the rigour of the computational approach (see below), which if addressed could help strengthen the conclusions of the study.
>> Thank you for your review. Your valuable feedback strengthens our work. We have addressed your major and minor concerns as outlined below:
- The use of the English language should be improved in this manuscript - it would benefit from a native speaker reading through. Also, since many of the results are of the speculative nature, the word "probably" in quite a few places ought to be softened to "may"/"can" (or perhaps "likely" e.g. line 178).
>> To address this issue, we invited Jacqueline Chyr, a native speaker, to fully revise and edit our manuscript. We have made many grammatical changes including replacing the word ‘probably’ with ‘may/can//likely /…’ as suggested.
- Chromosomes 11 and 19 are anyway highly gene-dense chromosomes (chr19 being the most gene-dense in the human genome). Are the editing-associated genes more enriched in these chromosomes than the background density?
>> Yes, editing-associated genes are more enriched in these chromosomes than the background density. To answer this question, we calculated the ratio of the number of editing-associated genes and the number of all the genes across the genomic regions. This analysis reduced the background noise. After that, these two regions (Chr11: 60M-70M and Chr19: 10M-20M) still have high density. We have added this density plot in the Figure S1.
- "1,146 miRNA-competing genes (Figure 4A, Table S7). Among these potentially affected genes, 127 are associated with Parkinson's disease" Here, the GO term from 4B is only based on those PD genes, so it shows that PD genes have PD-associated GO terms, which, if this is indeed a correct interpretation, isn't surprising. It does not show that PD genes are enriched in this 1,146 miRNA competing gene set. Therefore: Is there a significant overlap between the 127 PD genes and the miRNA competing genes?
>> Yes, there is a significant overlap between the 127 PD genes and the miRNA competing genes. We performed an enrichment analysis on all the 1,146 genes. The results show that these genes are enriched in PD-related biological processes, such as FoxO signaling pathway and neurotrophin signaling pathway. These processes are also the enriched pathways of the 127 PD genes. To present the whole enriched pathway of genes related to editing-mediated miRNA competitions, we have replaced the original Figure 4B with the new one, as suggested.
- Please expand on this "Of them, 18 RNAs have the relationships with both this editing event and its host gene (Figure 3D)" In 3D what are the bold genes? Furthermore, if you took random sets of 91 (non-EIF2AK2-target) genes, how many would also have this relationship? If it is fewer, then you are supporting the role of EIF2AK2-editing on its RBP function.
>> First question (In 3D what are the hold genes?): The bold genes are genes that are known to be implicated in PD. We have made this clarification in the main text and also the figure legend.
Second question (Furthermore, if you took random sets of 91 (non-EIF2AK2-target) genes, how many would also have this relationship?): To answer this, we performed the same analysis on all the genes (60,659) for the comparisons with the result of 91 genes. Of 60,659 genes, 7.11% have associations with both this editing event and EIF2AK2 (Table S6). Of 91 EIF2AK2-interacted protein-coding genes, 16 RNAs (16/91=17.58%) are significantly correlated to the editing event and EIF2AK2 (Figure 3D). There is an enrichment of EIF2AK2-interacted genes in the list of all the genes associated with the editing event. This suggests that EIF2AK2-editing may regulate these genes through its RNA binding functions. We have added this analysis in the third paragraph of Section 3.2.
- From the resource perspective it would be useful to know how shared are the editing sites across the different people in general? How much editing is present in each individual overall and does this correlate with disease state?
>> First question (How shared are the editing sites across different people?): All the informative editing events discussed in this study were shared by at least 50 (non-missing value) PD patients, as shown in the Section 2.2.
Second question (How many editing events are present in each individual overall and does this correlate with disease state?): There is an average of 15,104 editing events in each individual. The number of editing events has no associations with disease severity (P>0.05 for H&Y stages, alpha-synuclein, amyloid β-protein, tau, and unified Parkinson's disease rating scale) and there are no significant differences between PD samples and healthy controls (P>0.05).
Second question (How much frequency of editing events is present in each individual overall and does this correlate with disease state?): The mean frequency of informative editing events is 0.34±0.02 for each individual. This information shows significant differences between PD samples and healthy controls (P=5.92E-03). We have added the analysis result in the first paragraph of Section 3.1.
- For the QTL analysis FDR is used - what multiple testing corrections were used for the Pearson's correlation analysis? Furthermore, the strategy for the Pearson analysis is not well explained.
>> We clarified our strategy for the Pearson’s correlation analysis in our methods and discussion. In this study, QTL analysis was first performed to show the overview of RNA editing effects on genes. The aim of Pearson correlation analysis is to confirm the significant associations identified with the QTL analysis.
We applied multiple testing corrections for Pearson’s correlation: Among the 627,064 significant pairs identified by QTL analysis, 620,840 also passed Pearson correlation tests (P < 0.05). Of them, 620,616 editing-gene associations were still significantly correlated after Benjamini-Hochberg corrections (P.adjust < 0.05). We have included this information in the third paragraph of Discussion section.
- Cross-checking with the other cohorts is interesting and looks promising - however this part is first presented in the Discussion, perhaps this would be better included as part of the results themselves.
>> According to the Reviewer’s suggestion, we have added the cross-checking contents with the other cohorts as the last section of Results (Section 3.5).
- Since this is blood samples it might be a good idea to frame in the introduction what the connection is between the blood and Parkinson's disease.
>> As the Reviewer’s suggestion, we have checked the literature and described the increasing evidence of blood biomarkers in PD. The descriptions are highlighted in the third paragraph of Introduction section.
- "To investigate this, we also used TargetScan and miRanda to detect miRNA-binding regions in genes competing with the edited genes." - how is a 'competing' gene defined in this study?
>> We have added the definition of ‘competing gene’ in the second paragraph of section 2.5, which is highlighted in red. We also described ‘competing genes’ in detail throughout the manuscript.
- 2D what is the scale of this figure? Please write it in the legend
>> We have added the scale in the legend, which is highlighted in red.
- "we recognized 72 RNA editing events which may create 11,370 new miRNA-binding targets and eliminate 815" - can one speculate how the presence of the editing events would lead to the creation of so many miRNA binding targets, and also such a high ratio of created vs eliminated?
>> First question (can one speculate how the presence of the editing events would lead to the creation of so many miRNA binding targets): Yes, the presence of editing events in miRNA targets affect many binding sites of miRNAs. The reason is that there are 2,654 miRNAs from miRBase, some of which may share similarities in their binding sites.
Second question (can one speculate such a high ratio of created vs eliminated?) Yes, the high ratio may reveal one possible function of RNA editing events to suppress gene expressions overall. It is supported by 77.81% negative associations in all the 624,325 editing-gene pairs. We have added the analyses in the second paragraph of Discussion section. Additionally, we also searched the same results of miRNA binding alteration caused by RNA editing events in nine regions of AD patients [1] and 33 cancer types [2]. The results also show more gained miRNA binding targets than the lost targets caused by RNA editing events.
[1] Sijia Wu, Mengyuan Yang, Pora Kim, Xiaobo Zhou, ADeditome provides the genomic landscape of A-to-I RNA editing in Alzheimer’s disease, Briefings in Bioinformatics, Volume 22, Issue 5, September 2021, bbaa384, https://doi.org/10.1093/bib/bbaa384
[2] Sijia Wu, Zhiwei Fan, Pora Kim, Liyu Huang, Xiaobo Zhou, The Integrative Studies on the Functional A-to-I RNA Editing Events in Human Cancers, Genomics, Proteomics & Bioinformatics, 2023.
- 204 - please do not use the word validate, since it is just a correlation. Perhaps use "supported" instead.
>> As the Reviewer’s suggestion, we have checked the whole manuscript and replaced the word ‘validate’ with ‘support’.
- 207 - "we could annotate one potential mechanism" >> "we could speculate one potential mechanism"
>> As the Reviewer’s suggestion, we have revised this sentence.
- What is "Informative Value" in the context of calling RNA-editing sites?
>> Informative RNA-editing sites are sites that are present in more than 50 PD patients and have a standard deviation of greater than 0.05. We have clarified this in Section 2.2.
- The authors mention H&Y stages a few times but don't mention what it is. Please say somewhere so that readers avoid having to search themselves
>> We have added the full name (Hoehn and Yahr staging scale) and function (described the severity of the disease) of H&Y stages in the Section 2.1.
- Figure 3 - the bright green and red is very distracting (and probably not great for color blind readers).
>> Thank you for this suggestion, we have altered the colors in Figure 3, Figure 4, Figure 5, Figure S2, and Figure S3.
- What is the measurement/scale in Figure S3?
>> We have added the measurement/scale of the expression values from different datasets in the legend of Figure S3.
- Since the focus was on the sites already present in the REDIportal, can the authors speculate on whether more PD-specific sites could be found from the data, which are not in the the portal, and the potential impact of this?
>> There may be more PD-related editing sites that are not in the portal. However, to increase the reliability of RNA editing sites and reduce the number of false-positives, we focused on the sites in REDIportal. Other studies [1-2] also focus on editing sites found in the portal.
[1] Riella C V, McNulty M, Ribas G T, et al. ADAR regulates APOL1 via A-to-I RNA editing by inhibition of MDA5 activation in a paradoxical biological circuit[J]. Proceedings of the National Academy of Sciences, 2022, 119(44): e2210150119.
[2] Lin C H, Chen S C C. The cancer editome atlas: a resource for exploratory analysis of the adenosine-to-inosine RNA editome in cancer[J]. Cancer Research, 2019, 79(11): 3001-3006.
Round 2
Reviewer 2 Report
The authors have addressed my comments in the revised version of their manuscript. I have a few hopefully very minor comments regarding their revisions:
Line 215: the authors have followed the suggestion about checking the background proportion, I have two comments though:
1. 60,659 seems like a lot of 'genes' (transcripts?) to include in the background - please state where this figure has come from as it's not clear to me. To make a fair comparison with EIF2AK2 targets (which are presumably expressed in blood) , the authors ought to only include genes sufficiently expressed in blood in the background set.
2. Please also include a p.value from using a fishers test or similar to show that the enrichment is significant (it is indeed using the numbers shown but I'm not convinced 60,659 genes is the best number to compare to).
Line 333: As per my suggestion the authors have moved some of the results based on other datasets from the Discussion to the results section. However these results appear to start from 340 and the first paragraph of this section looks more like Discussion text - can the authors please clear this text up a bit?
The English is much better, though it could still be helpful for the native speaker to go through one more time as I can still spot some errors.
The authors included the significant differences in editing frequency (line 159) but I can't see that they included the reply comment about the actual number of editing events having no significant relationship with the disease - perhaps a short mention would be interesting?
One last thing: in the supplementary tables (and associated website linking from them) it appears the authors have importantly put efforts into making the resource available with lots of tables. However, wouldn't it also be useful for biologists if they also include a list of their original 25,799 detected editing sites in their Excel sheet?
Author Response
The authors have addressed my comments in the revised version of their manuscript. I have a few hopefully very minor comments regarding their revisions:
>> Thank you for your review. We have addressed your concerns as outlined below:
- Line 215: the authors have followed the suggestion about checking the background proportion, I have two comments though: (1). 60,659 seems like a lot of 'genes' (transcripts?) to include in the background - please state where this figure has come from as it's not clear to me. To make a fair comparison with EIF2AK2 targets (which are presumably expressed in blood), the authors ought to only include genes sufficiently expressed in blood in the background set. (2) Please also include a p.value from using a fishers test or similar to show that the enrichment is significant (it is indeed using the numbers shown but I'm not convinced 60,659 genes is the best number to compare to).
>> The first question: The 60,659 genes are all the genes identified from RNAseq data without any filtering process.
The second question: According to the Reviewer’s suggestion, we then only consider 6,287 genes sufficiently expressed in blood (mean expression > one TPM). To define the potential EIF2AK2 gene targets, we collect both predicted (from RNAInter and RNAct) and experimentally validated (from StarBase) EIF2AK2-RNA interactions. Of the 6,287 genes, 2,158 genes are predicted or identified as EIF2AK2 targets and other 4,129 genes are non EIF2AK2 targets. Of the 2,158 potential EIF2AK2 targets, 31.37% genes are significantly associated with EIF2AK2 and the editing event. Of the 4,129 potential non EIF2AK2 targets, 26.45% genes are significantly associated with EIF2AK2 and the editing event. There is an enrichment of EIF2AK2-interacted genes associated with the editing event (Fisher test: P = 4.54E-05). We have added the analyses in the third paragraph of Section 3.2.
- Line 333: As per my suggestion the authors have moved some of the results based on other datasets from the Discussion to the results section. However these results appear to start from 340 and the first paragraph of this section looks more like Discussion text - can the authors please clear this text up a bit?
>> According to the suggestion, we have removed the first paragraph of Section 3.5 to the second paragraph of Discussion.
- The English is much better, though it could still be helpful for the native speaker to go through one more time as I can still spot some errors.
>> According to the suggestion, Jacqueline Chyr has checked the manuscript one more time. The revisions are highlighted in red.
- The authors included the significant differences in editing frequency (line 159) but I can't see that they included the reply comment about the actual number of editing events having no significant relationship with the disease - perhaps a short mention would be interesting?
>> According to the suggestion, we have added the description about the number of editing events in the Section 2.2.
- One last thing: in the supplementary tables (and associated website linking from them) it appears the authors have importantly put efforts into making the resource available with lots of tables. However, wouldn't it also be useful for biologists if they also include a list of their original 25,799 detected editing sites in their Excel sheet?
>> According to the suggestion, we have included the identified RNA editing events in the website (https://www.synapse.org/#!Synapse:syn39828010/files/) shown in Table S3.